# SNARE protein USE1 is involved in the glycosylation and the expression of mumps virus fusion protein and important for viral propagation

**Yaqing Liu[1,2]◉, Hiroshi Katoh◉[1◉¤]\*, Tsuyoshi Sekizuka[3], Chaewon Bae[1], Aika Wakata[1], Fumihiro Kato[1], Masafumi Sakata[1], Toshiyuki Yamaji[4], Zhiyu Wang[2], Makoto Takeda[1¤]**

**1** Department of Virology III, National Institute of Infectious Diseases, Musashimurayama, Tokyo, Japan, **2** Department of Virology, School of Public Health, Cheeloo College of Medicine, Shandong University, Jinan, Shandong, China, **3** Pathogen Genomics Center, National Institute of Infectious Diseases, Shinjuku-ku, Tokyo, Japan, **4** Department of Biochemistry and Cell Biology, National Institute of Infectious Diseases, Shinjuku-ku, Tokyo, Japan

◉ These authors contributed equally to this work.

¤ Current address: Department of Microbiology, Graduate School of Medicine and Faculty of Medicine, The University of Tokyo, Bunkyo-ku, Tokyo, Japan

\* hirokato@m.u-tokyo.ac.jp

**Data Availability Statement:** All relevant data are within the manuscript and its Supporting Information files. MS proteomics data have been

## Abstract

Mumps virus (MuV) is the etiological agent of mumps, a disease characterized by painful swelling of the parotid glands and often accompanied by severe complications. To understand the molecular mechanism of MuV infection, a functional analysis of the involved host factors is required. However, little is known about the host factors involved in MuV infection, especially those involved in the late stage of infection. Here, we identified 638 host proteins that have close proximity to MuV glycoproteins, which are a major component of the viral particles, by proximity labeling and examined comprehensive protein–protein interaction networks of the host proteins. From siRNA screening and immunoprecipitation results, we found that a SNARE subfamily protein, USE1, bound specifically to the MuV fusion (F) protein and was important for MuV propagation. In addition, USE1 plays a role in complete N-linked glycosylation and expression of the MuV F protein.

## Author summary

The family *Paramyxoviridae* includes several important human and animal pathogens such as mumps virus (MuV), which is the etiological agent of mumps. Viral particle formation, the final stage of viral infection, is a complex and integrated step in which all of the particle components accumulate and assemble. To understand the molecular mechanism of this stage of MuV infection, we searched for host proteins that came into close proximity to the cytoplasmic domain of viral membrane proteins, which are a major viral particle component. We generated comprehensive protein–protein interaction networks of the obtained host proteins and found that a SNARE subfamily protein, USE1, was

deposited to the ProteomeXchange Consortium (PXD035151).

**Funding:** This work was supported by grants from the Japan Agency for Medical Research and Development (AMED) (the Japan Program for Infectious Diseases Research and Infrastructure, JP21wm0325024j0002 and the Research Program on Emerging and Re-emerging Infectious Diseases, JP21fk0108623j0001, JP21fk0108617j0201, and JP21fk0108087j0403) and the Japanese Society for the Promotion of Science (JSPS) (Grant-in-Aid for Scientific Research (C), 19K07584) to HK. The funders had no role in study design, data collection and analysis, decision to publish, or preparation of the manuscript.

**Competing interests:** The authors have declared that no competing interests exist.

important for the efficient MuV propagation. Furthermore, this protein bound specifically to the MuV F protein and was important for the functional maturation of F protein through complete N-linked glycosylation and expression.

## Introduction

Mumps is a common childhood illness characterized by painful swelling of the parotid glands and is often accompanied by severe complications, such as orchitis, aseptic meningitis, pancreatitis, and deafness [1]. *Mumps orthorubulavirus* (MuV), which belongs to the genus *Orthorubulavirus* within the family Paramyxoviridae, is the causative agent of mumps [2]. Infection by this virus is initiated by the binding of the hemagglutinin-neuraminidase (HN) protein to sialic acids of the host cell surface [3]. After receptor binding, the fusion (F) protein induces pH-independent fusion of the viral envelope with the host plasma membrane, and the viral genomic RNA is released into the cytoplasm [4]. The viral genomic RNA encapsidated by the nucleocapsid (N) protein forms an active template for RNA replication and transcription, a viral ribonucleoprotein (vRNP), with viral polymerases composed of the phospho-(P) and large (L) proteins [5]. Viral structural components synthesized in the cytoplasm are transported to the plasma membrane (PM). At the PM, the matrix (M) protein organizes the assembly of vRNP complexes (composed of N, P, L, and genomic RNA) and envelope proteins (F and HN), leading to the efficient budding and release of progeny virions from infected cells [6].

MuV F protein is a trimeric type I membrane protein that is synthesized as an inactive precursor ($F_0$) in the endoplasmic reticulum (ER) [7]. After being glycosylated in the Golgi, it is transported to the trans-Golgi network (TGN) and cleaved by an endoprotease, furin, into the disulfide-linked subunits $F_1$ and $F_2$ [8]. The complex of trimeric F and tetrameric HN proteins causes cell–cell and virus–cell fusion in concert [9]. Specific N-glycans on the F proteins of several paramyxoviruses, such as Newcastle disease virus (NDV), measles virus (MeV), and Sendai virus (SeV), are important for this fusion activity [10–12], whereas N-glycans on the Nipah virus (NiV) F protein reduce cell–cell fusion [13]. N-glycans on many viral glycoproteins are required for protein stability, intracellular transport, proteolytic activation, protection from proteolytic degradation, and escape from antibody recognition [13–16]. N-linked glycosylation is a conserved process in all proteins that undergo this modification, and although many of the host enzymes involved in this process have been identified, further exploration of key host factors and analysis of their functions are needed to better understand the mechanism of glycosylation of paramyxovirus F proteins [17].

Proximity labeling (PL) is an alternative to traditional affinity purification methods, such as immunoprecipitation, for conducting a proteomic analysis of protein interaction networks [18]. In PL, promiscuous enzymes are genetically targeted to the protein complex of interest. The addition of a biotin-derived small molecule substrate initiates the covalent tagging of endogenous proteins located within a few nanometers of the promiscuous enzymes. Subsequently, the biotinylated proteins are harvested using streptavidin and identified by mass spectrometry (MS). Therefore, PL is a valuable tool for studying the spatial and interaction characteristics of proteins in living cells. A biotin ligase, TurboID, has been shown to catalyze PL with a level of efficiency much greater than that of previously used enzymes such as APEX2 and BioID [19]. Recently, split-TurboID, which is a pair of inactive TurboID fragments, was reported [20]. Split-TurboID can be conditionally reconstituted for performing spatially specific PL, especially PL at organelle contact sites or macromolecular complexes, in cells. In

addition, split-TurboID may be applicable for use with proteins in which the TurboID gene cannot be inserted owing to their molecular size or structure.

Here, we comprehensively identified host proteins in close proximity to MuV membrane proteins by using split-TurboID, generated protein–protein interaction (PPI) networks, and analyzed the function of a soluble N-ethylmaleimide-sensitive factor attachment protein receptor (SNARE) subfamily protein, Unconventional SNARE in ER 1 (USE1) during MuV infection.

## Results

### Introduction of split-TurboID into the cytoplasmic region of MuV glycoproteins to search for host factors involved in the late stages of MuV infection

Because efficient particle formation for MuV requires, at minimum, the MuV N, M, and F proteins, we initially attempted to introduce the biotin ligase TurboID into any one of these proteins [21]. However, no region was identified in any of these proteins that satisfied the requirements for allowing both TurboID insertion and virus-like particle (VLP) production (S1 Fig shows an example result when TurboID was inserted into the C-terminus of the M protein). We next tried to introduce a split-TurboID instead, and we succeeded in introducing the N-terminal and C-terminal fragments of split-TurboID (Tb(N) and Tb(C)) into the cytoplasmic domains of the MuV HN and F proteins, respectively (Fig 1A) [20]. Regarding its effect on VLP production, the insertion of split-TurboID was tolerated, but it reduced the level of VLP production by ~10-fold (Fig 1B and 1C). The interactions among the M, F-FLAG-Tb (C), and Tb(N)-HA-HN proteins were confirmed by immunoprecipitation (Fig 1D). To investigate the efficiency and specificity of split-TurboID-mediated biotinylation, we assessed biotinylated proteins in cells expressing either or both F-FLAG-Tb(C) and Tb(N)-HA-HN proteins in the presence or absence of biotin treatment. Elevated amounts of biotinylated proteins were observed only in the cells that expressed both the F-FLAG-Tb(C) and Tb(N)-HA-HN proteins and had been treated with biotin (Fig 1E). In addition, self-biotinylation of the F-FLAG-Tb(C) and Tb(N)-HA-HN proteins was confirmed, demonstrating the successful reconstitution and functional ligase activity of split-TurboID (Fig 1F). Immunofluorescence imaging revealed that biotinylated proteins were co-localized with the M, F, and HN proteins in the vicinity of the plasma membrane (Fig 1G).

### Interactome analysis of host proteins in close proximity to MuV glycoproteins

Following affinity purification and MS, we calculated the average of the detection intensity (protein abundance) obtained from each sample (control and TurboID), extracted factors that were detected only in the split-TurboID sample (486 proteins) or factors with an intensity ratio (TurboID/control) of ≥10-fold (160 proteins), and identified a total of 641 proteins, including the M, F-FLAG-Tb(C), and Tb(N)-HA-HN proteins, as being significantly enriched in the split-TurboID-expressing cells (S2 Fig and S1 Table). To categorize functionally related proteins, a gene ontology (GO) analysis was performed. The significant terms were selected by applying the thresholds of a $p$-value of <0.05, re-ranked false discovery rate (RFDR) of <0.05, and fold enrichment of ≥5. Among the 638 host proteins, 277 proteins were categorized into four broad functional clusters: cell adhesion, nuclear-related protein, translation, and vesicle transport (32 GO biological process terms, 9 GO molecular function terms, and 19 GO cellular component terms) (Fig 2A).

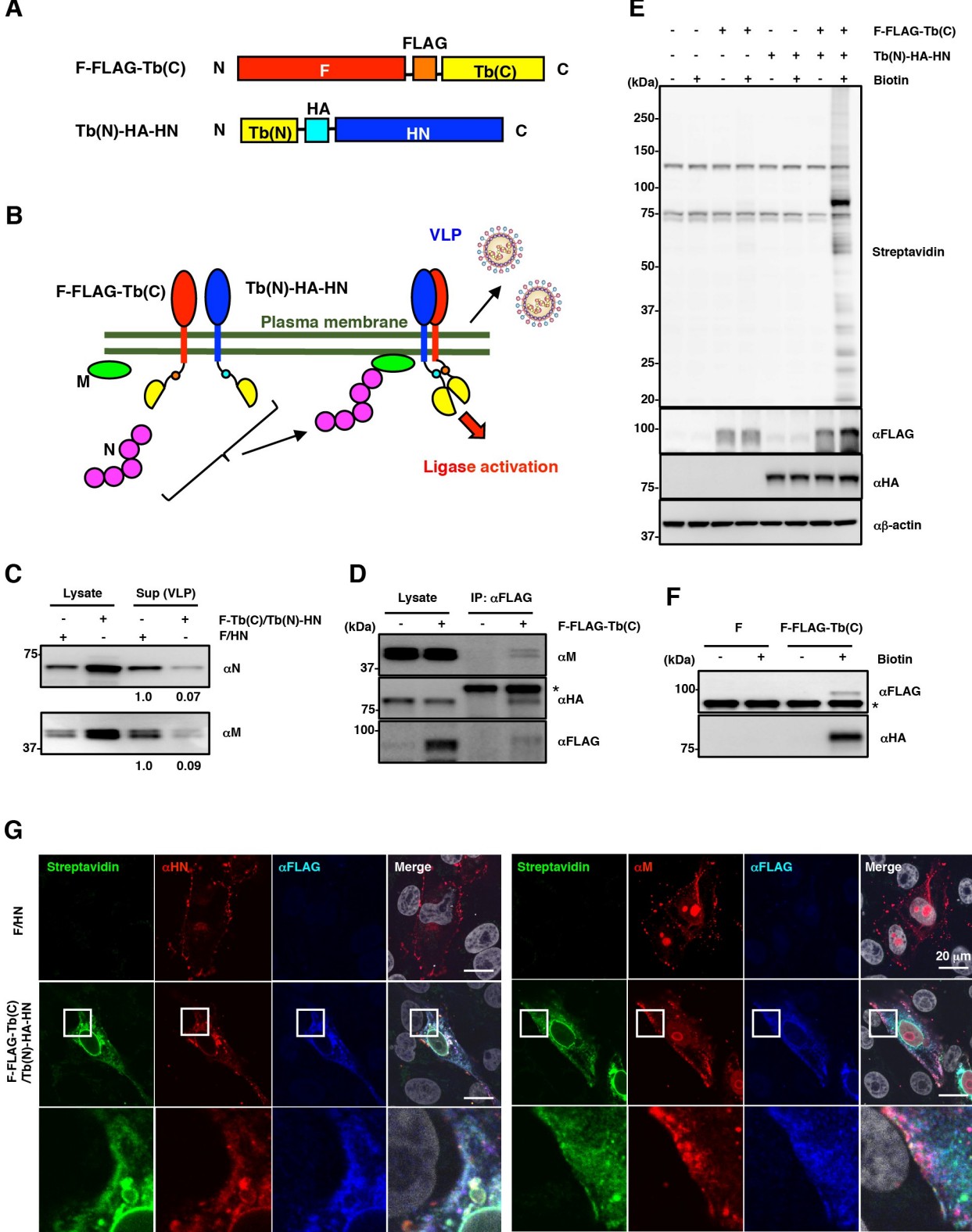

**Fig 1. Search for host factors located in the vicinity of the cytoplasmic tails of the MuV F and HN proteins. (A)** Schematic diagrams of F-FLAG-Tb(C) and Tb(N)-HA-HN. **(B)** Schematic overview of the split-TurboID-based proximity biotinylation using a virus-like particle (VLP) assay system. **(C)** VLP assay showing the amounts of the N and M proteins expressed in the cells (Lysate) and released into the supernatants (Sup).

**(D)** Immunoprecipitation assay showing the interaction of F-FLAG-Tb(C) protein with Tb(N)-HA-HN and M proteins in 293T cells. **(E)** Immunoblotting showing the ligase activity of split-TurboID inserted into the cytoplasmic domains of MuV F and HN proteins in 293T cells. **(F)** Immunoblotting showing the biotinylated F-FLAG-Tb(C) and Tb(N)-HA-HN proteins in 293T cells. **(G)** Immunofluorescence assay showing the biotinylated proteins in Vero cells expressing F-FLAG-Tb(C) and Tb(N)-HA-HN proteins. The experiments were performed at least three times, and representative data are shown.

To examine the PPI networks of each factor extracted by the GO analysis, a Search Tool for the Retrieval of Interacting Genes (STRING) network analysis was performed (Fig 2B). Although "cell adhesion" had a high score in the GO analysis, this high score is likely a consequence of a typical limitation of gene annotation databases because many genes play multiple roles in numerous cellular functions, and a small cluster (cluster I in Fig 2B) was formed from only a limited number of factors belonging to this term. Many factors involved in "vesicle transport", such as vesicle-mediated transport between the ER, Golgi, and endosomes, vesicle fusion, and vesicle organization, were enriched in the area with close proximity to the cytoplasmic region of MuV glycoproteins and formed a PPI network (cluster II in Fig 2B). Cluster III was composed of proteins categorized as "nuclear-related protein". Because the wildtype viral proteins were not localized in the nuclear membrane (Fig 1G), the detection of these nuclear-related proteins might be an artifact of the addition of split-TurboID to the cytoplasmic region of the glycoproteins. No distinct cluster was formed by the factors belonging to "translation", including components of the eIF3 and eIF4 complexes. Because no or only artefactual clusters in the STRING network analysis were observed for "cell adhesion", "nuclear-related protein", and "translation" factors, whereas the "vesicle transport" factors formed a large cluster, we focused on the host proteins identified by split-TurboID-based PL that were categorized as being related to "vesicle transport" and further investigated their roles in MuV propagation.

## Selection by siRNA screening of host factors required for MuV propagation

To identify host factors that are essential for MuV propagation, we used the recombinant MuV expressing an *Aequorea coerulescens* green fluorescent protein (rMuV/AcGFP)-based screening method described previously [22]. From the detection intensity in a mass spectrometry analysis, we selected 24 of the proteins categorized as being related to "vesicle transport". siRNA-treated A549 cells were infected with rMuV/AcGFP, and viral growth was assessed by the level of AcGFP expression (Fig 3A). Among the 24 tested factors, the 13 factors that showed a reproducible and significant decrease in AcGFP expression were further investigated. A549 and Vero cells treated with each siRNA were infected with rMuV/AcGFP, and the viral titers in culture supernatants were measured. As shown in Fig 3B, knockdown of all factors except SEC62 significantly reduced viral production in A549 cells, correlating with the result of reduced AcGFP signaling. In contrast, suppression of viral growth in Vero cells was observed only in those transfected with one of six siRNAs (BET1, EMD, LRRC59, LYPLA2, USE1, and VAPA), and the level of viral growth suppression was weaker than that in correspondingly treated A549 cells. Among those six factors, siLYPLA2 treatment was cytotoxic to Vero cells (Fig 3C). From these results, five factors (BET1, EMD, LRRC59, USE1, and VAPA) were selected as candidate host proteins important for MuV propagation.

## BET1 and USE1 are critical host factors for MuV propagation through their interaction with the F protein

To examine the interaction of MuV F and/or HN proteins with each target host protein, FLAG-tagged F or HN protein and an HA-tagged host protein (HA-BET1, HA-EMD,

**A**

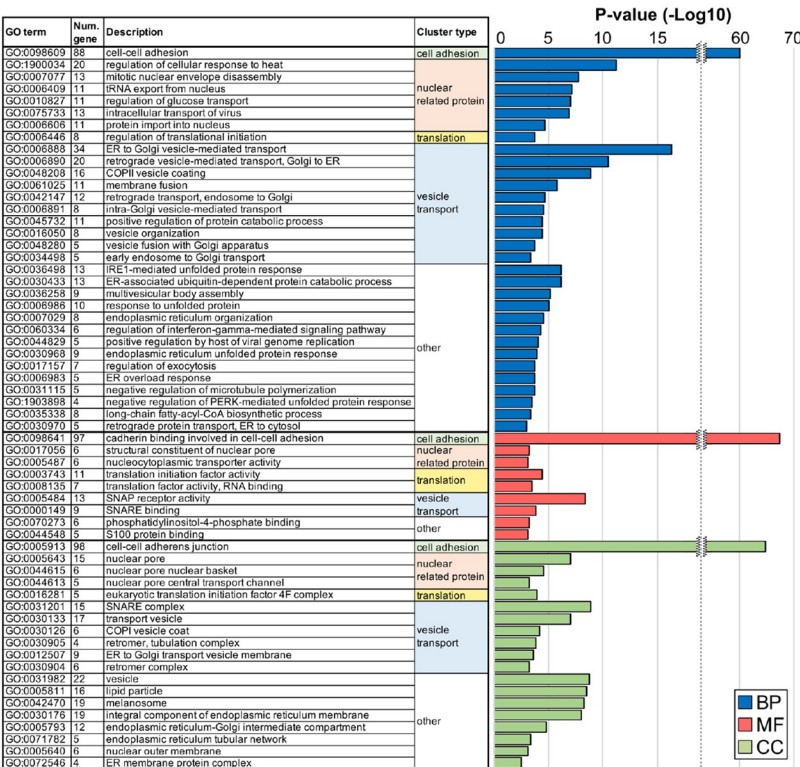

**B**

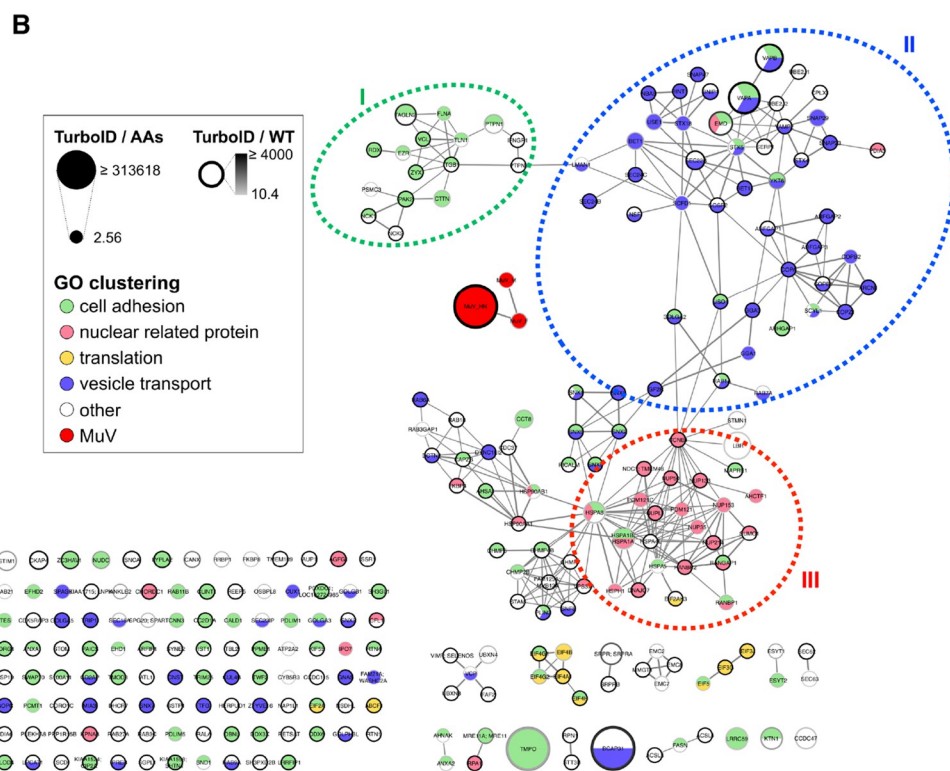

**Fig 2. Functional classification of host factors in close proximity to the MuV F and HN proteins.** (**A**) Sixty gene ontology (GO) terms were found to be significant (*p*-value < 0.05, RFDR < 0.05, and fold enrichment ≥ 5) and were assigned to four broad functional categories: cell adhesion, nuclear-related protein, translation, and vesicle transport. BP: biological processes, MF: molecular function, CC: cellular components. (**B**) PPI networks of the obtained factors from the GO analysis were illustrated by performing a STRING network analysis. The nodes represent each factor, and the edges represent interactions. The node size represents the protein abundance corrected for the number of amino acid residues (AAs), and the density of the node outline represents the intensity ratio (protein abundance in the TurboID sample/protein abundance in the wildtype [WT] sample).

HA-LRRC59, HA-USE1, or VAPA-HA) were co-expressed and immunoprecipitated with anti-FLAG or anti-HA antibody. HA-BET1 and HA-USE1 co-precipitated with the F-FLAG protein but not with the FLAG-HN protein (Fig 4A). VAPA-HA interacted weakly with both the F-FLAG and FLAG-HN proteins (S3 Fig). No interactions between the viral glycoproteins and the other tested host proteins were observed. Thus, we focused on BET1 and USE1 as candidate host proteins of importance in the late stages of the MuV lifecycle.

Both BET1 and USE1 are the members of the SNARE protein family and function in COPII- and COPI-mediated vesicle transport between the ER and Golgi [23]. BET1 localized close to Calnexin, an ER marker protein (S4A Fig). Although the anti-USE1 antibody was not used for the detection of an endogenous USE1 protein by immunofluorescence assay, it successfully recognized HA-tagged USE1 (S4B Fig). Because HA-tagged USE1 co-localized with Calreticulin, another ER marker protein, endogenous USE1 was presumed to be located in the ER as well (S4C Fig). After translation, the MuV F protein was transported to the PM via the secretory pathway. Up to 18 h post-infection (hpi), the F protein was primarily localized in the cytoplasm, but by 24 hpi, most F proteins had reached the PM. From the immunofluorescence assay results, the F protein was proximal to or partially co-localized with USE1 in the ER at 12–18 hpi, whereas the co-localization of F protein and BET1 was not clear (Figs 4B and 4C and S5).

Next, we confirmed the importance of BET1 and USE1 for MuV propagation. A549 cells treated with siRNA directed against BET1 or USE1 or with negative control siRNA (siBET1, siUSE1, or siNC, respectively) were infected with rMuV/AcGFP at a multiplicity of infection (MOI) of 0.05. Although the levels of viral RNA (both mRNA and genomic RNA) and protein expression in BET1-knockdown cells were comparable until 96 hpi, the final virus titer was lower than that of control cells at 96 hpi (Fig 5). In USE1-knockdown cells, the levels of viral RNA and proteins were all suppressed after 48 hpi, and the titers of infectious MuV were lower than those of control cells at all analyzed timepoints. These findings indicate that BET1 and USE1 interact with the MuV F protein in the ER and are important for MuV propagation.

## USE1 is involved in the glycosylation and expression of MuV F protein

To investigate the effect of BET1 or USE1 knockdown on viral fusion activity, the F protein was expressed along with the HN protein and AcGFP. Syncytium formation was strongly inhibited in the USE1-knockdown A549 cells but was only slightly inhibited in the BET1-knockdown A549 cells (Fig 6A). Because siUSE1 had stronger effects compared with siBET1, subsequent investigations were conducted using siUSE1. To confirm the reduction in fusion activity induced by siUSE1, a dual split protein (DSP)-based fusion assay was performed. This assay uses two DSPs, which are chimeras of split forms of GFP and renilla luciferase (RL) ($DSP_{1-7}$ and $DSP_{8-11}$) [24]. The fusion of two cell lines expressing the respective DSP induced by the F and HN proteins results in the fusion and reconstitution of GFP and RL. As shown in Fig 6B, GFP expression and RL activity were significantly suppressed by USE1-knockdown.

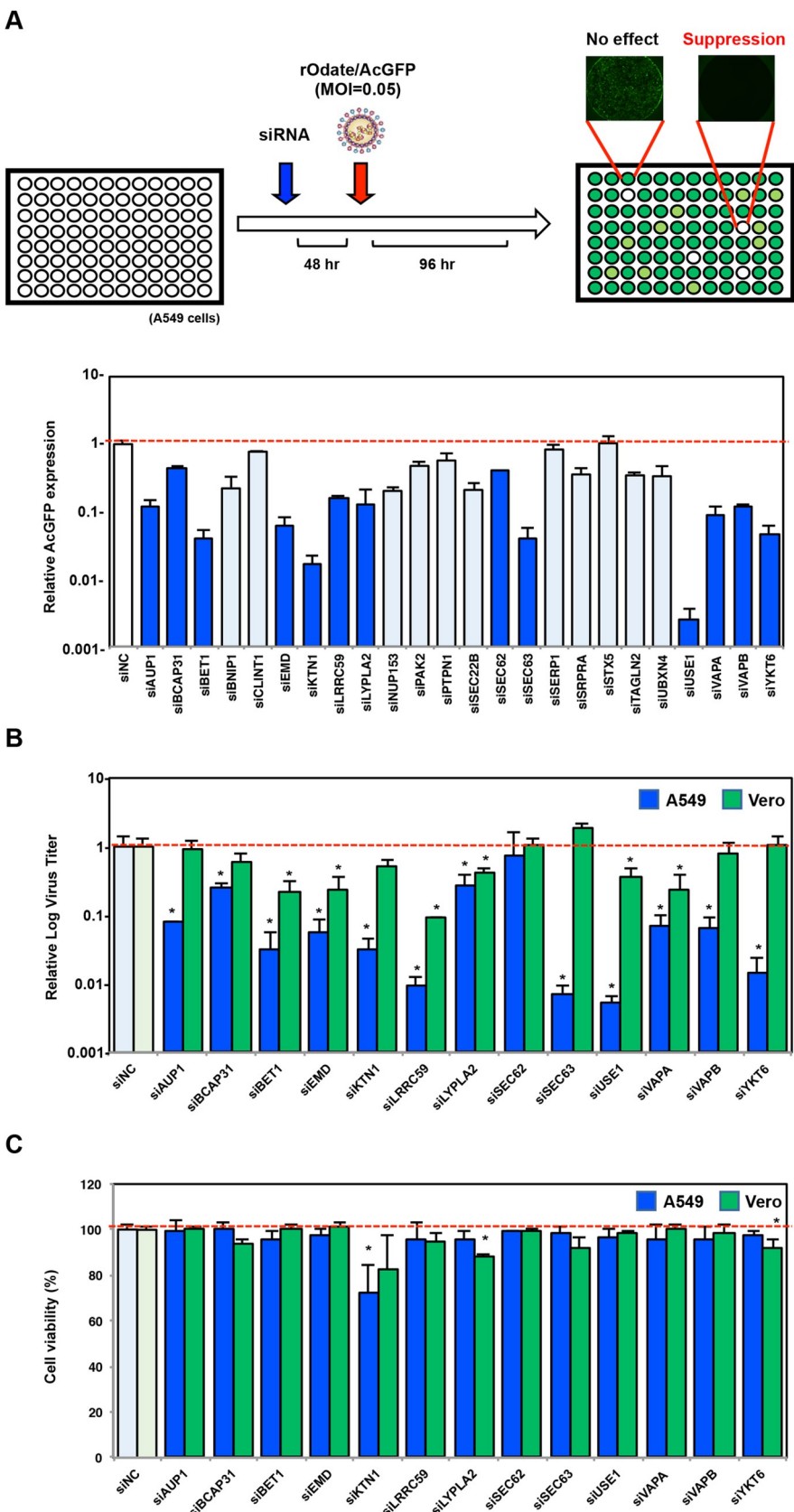

**Fig 3. Identification of host proteins required for MuV propagation. (A)** Schematic overview of the rMuV/AcGFP-based siRNA screening system (Upper). Relative AcGFP expressions in rMuV/AcGFP-infected A549 cells transfected with the indicated siRNA (Lower). Error bars show the standard deviations of triplicate wells. The 13 factors used in the further experiments whose data are displayed in **(B)** and **(C)** are shown in dark blue bars. **(B)** Relative virus titers in A549 or Vero cells transfected with the indicated siRNA. Error bars show the standard deviations of triplicate wells. Differences with a *p*-value of <0.01(**) were considered to be statistically significant. (n.s. = not significant) **(C)** Relative cell viability of A549 or Vero cells transfected with the indicated siRNA. Error bars show the standard deviations of triplicate wells. Differences with a *p*-value of <0.01(**) were considered to be statistically significant. (n.s. = not significant).

We examined the effects of USE1-knockdown on the expression of the F and HN proteins by SDS-PAGE and immunoblotting. A single band corresponding to the F-FLAG protein was observed in control 293T cells, whereas two bands with fast mobility were observed in USE1-knockdown cells (Fig 6C). To determine whether the band observed in control cells was uncleaved $F_0$ or cleaved $F_1$ protein, the Fcm protein with mutations in the furin cleavage motif was expressed in 293T cells. As shown in Fig 6D, the ~75-kDa band corresponding to the uncleaved $F_0$ protein was detected in control cells. Therefore, the ~60-kDa band detected in the cells expressing the wildtype F protein was considered to be cleaved $F_1$ protein. In contrast to the effect of USE1 knockdown on the F protein, USE1 knockdown did not change the expression pattern of the HN protein.

N-linked glycans attach to paramyxoviral F protein and contribute to its functions. Therefore, we next investigated whether the change in the mobility of the F protein bands was due to a change in the glycan chains added to the F protein, using two endoglycosidases, Endo H and PNGase F. Endo H does not cleave complex type N-glycans, which are acquired in the medial- and trans-Golgi, whereas PNGase F eliminates all N-linked oligosaccharide chains. The band pattern of the non-glycosylated F protein in the cells treated with PNGase F was similar regardless of USE1 knockdown, suggesting that the difference observed in the USE1-knockdown cells was due to changes in N-linked glycosylation (Fig 6E). In the Endo H-treated control cells, several bands of lower molecular weights were observed, indicating that both high-mannose and hybrid/complex type glycans were attached to the MuV F protein. However, in USE1-knockdown cells, Endo H treatment caused the upper band to disappear and resulted in an increase in the intensity of the lower band. These data also demonstrate that USE1 is involved in the N-linked glycosylation on MuV F protein.

To further confirm the importance of USE1 for F glycosylation, we used MGAT1 knockout (KO) HeLa cells, which lack hybrid/complex type glycans [25]. As shown in Fig 6F, the molecular size of the F-FLAG protein in USE1-knockdown HeLa/Parent cells was smaller than that in control HeLa/Parent cells. In contrast, a similar F-FLAG protein band pattern was observed in both the control and USE1-knockdown HeLa/MGAT1 KO cells. The molecular weight of glycosylated integrin β1 was also changed by USE1 knockdown only in HeLa/Parent cells (Fig 6F). These data show that USE1 is essential for complex type N-linked glycosylation on several membrane proteins, including both viral (MuV F) and cellular (integrin β1) proteins. Next, the cell surface expression of F protein was analyzed using a biotinylation approach. The transfected 293T cells were labeled with a biotinylation reagent that reacts with only cell surface proteins because the reagent is not PM permeable, and then subjected to lysis and immunoprecipitation. Total and cell surface-expressed F proteins were detected by anti-FLAG antibody and streptavidin, respectively. As shown in Fig 6G, the expression level of F protein on the surface of USE1-knockdown cells was comparable with that of control cells, indicating that the knockdown of USE1 did not affect cell surface transport of the F protein.

Finally, we investigated the effects of USE1 knockdown on the F glycosylation in A549 and Vero cells. As shown in Fig 6H, the expression of F protein in A549 cells was significantly

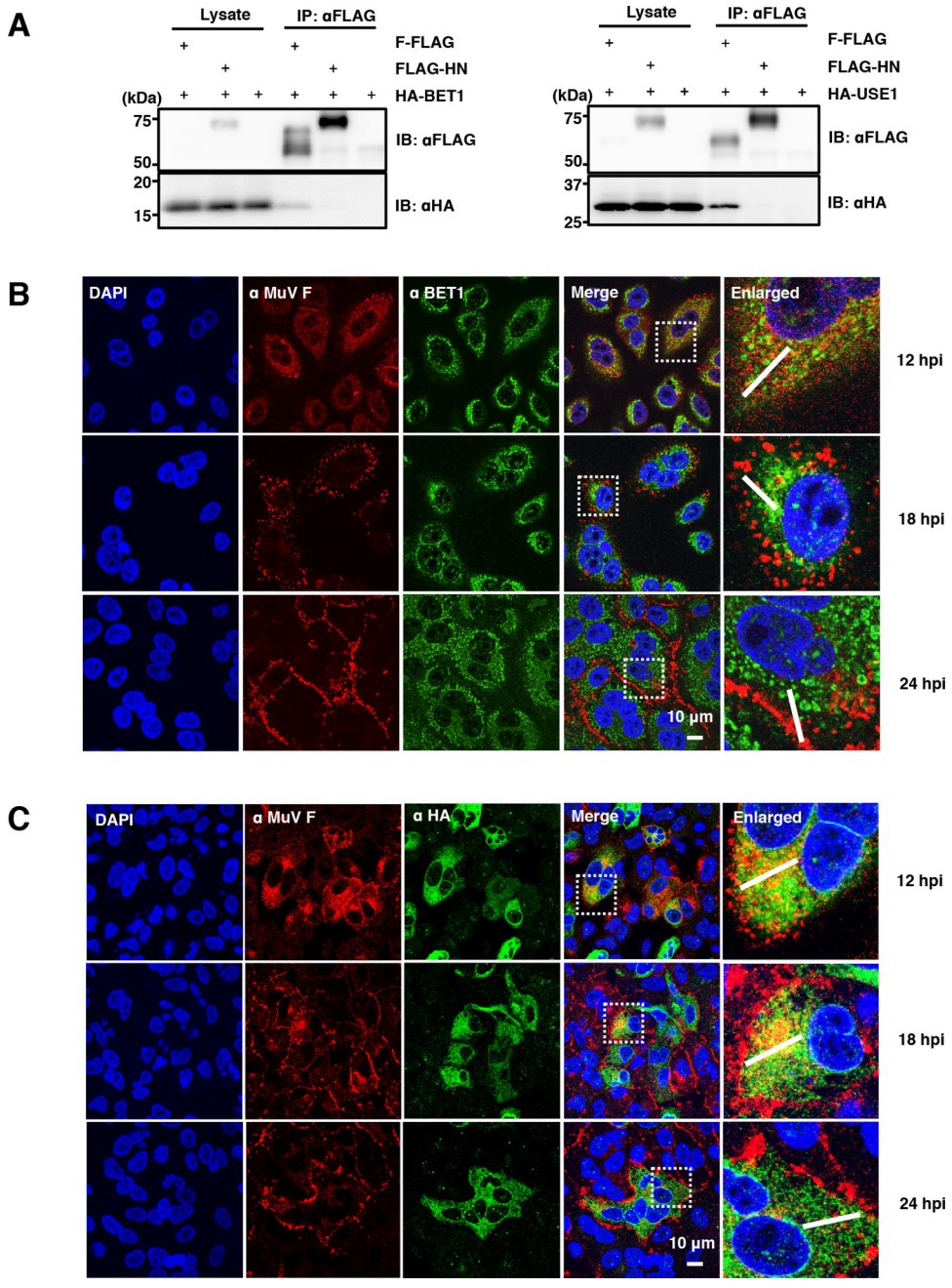

**Fig 4. Interaction of BET1 and USE1 with the MuV F protein in the ER.** (**A**) Immunoprecipitation assay showing the interaction of F-FLAG or FLAG-HN proteins with HA-BET1 (left) or HA-USE1 (right) in 293T cells. (**B**) Immunofluorescence assay showing the localization of MuV F protein and BET1 in A549 cells. (**C**) Immunofluorescence assay showing the localization of MuV F protein and USE1 in A549 cells expressing HA-USE1. The experiments were performed at least three times, and representative data are shown.

decreased by USE1 knockdown, whereas the expression level and the glycosylation of F protein in Vero cells were not affected by USE1-knockdown. Taken together, these data indicate that USE1 is involved in the maturation of the F protein, including its glycosylation and expression, but USE1 function is cell-type dependent.

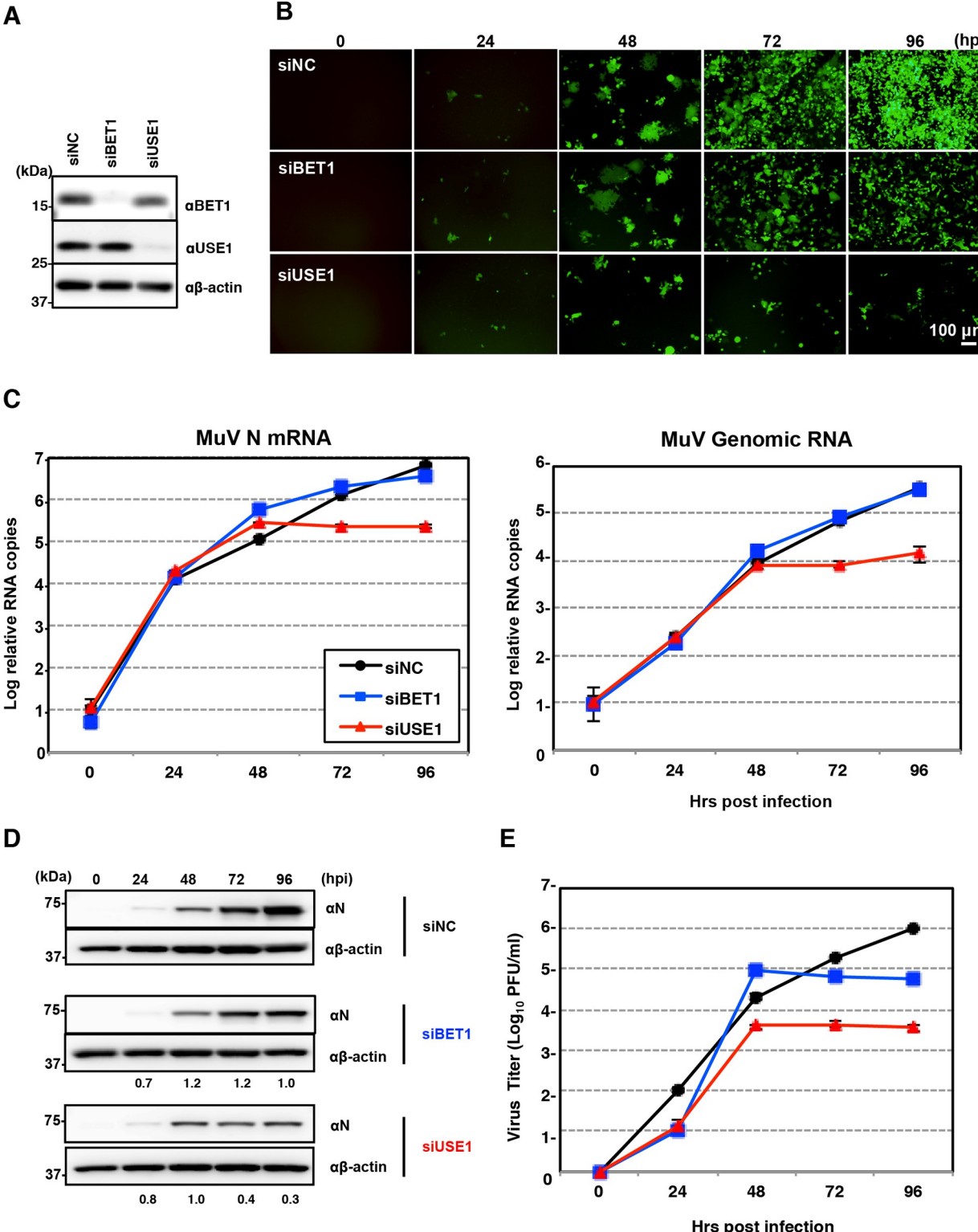

**Fig 5. Importance of BET1 and USE1 for MuV propagation. (A)** Immunoblotting showing the expression levels of BET1, USE1, and β-actin in A549 cells transfected with siBET1, siUSE1, or siNC. **(B)** Immunofluorescence of rOdate/AcGFP-infected A549 cells transfected with siBET1, siUSE1, or siNC at the indicated times. **(C)** Relative expressions of viral mRNA and genomic RNA in A549 cells transfected with siBET1 (blue line), siUSE1 (red line), or siNC (black line). **(D)** Immunoblotting showing the expression levels of MuV N protein and β-actin in A549 cells transfected with siBET1, siUSE1, or siNC. The relative band intensities of MuV N protein in A549 cells transfected with siBET1 or siUSE1 were

normalized by those in cells transfected with siNC. **(E)** Growth kinetics of rMuV/AcGFP in A549 cells transfected with siBET1 (blue line), siUSE1 (red line), or siNC (black line). Error bars indicate the standard deviations of triplicate wells. The experiments were performed at least three times, and representative data are shown.

## Discussion

Here, we identified the PPI networks of the host proteins in close proximity to the cytoplasmic region of MuV glycoproteins. Among the host proteins, a SNARE subfamily protein, USE1, was found to be important for efficient MuV propagation. Furthermore, we determined that USE1 bound specifically to the MuV F protein and was involved in the N-linked glycosylation and the expression of F protein.

We used TurboID in the present study to illustrate the protein composition of MuV F–HN contact sites [19]. Because the TurboID biotin ligase biotinylates proteins that are in close proximity to it, the affected proteins do not necessarily have a tight or prolonged interaction with the protein to which TurboID is connected; thus, the identified proteins may interact with the MuV glycoproteins tightly or weakly, and they may come into close proximity of the cytoplasmic region of these glycoproteins transiently or remain there for long periods. Therefore, the present data could be considered a comprehensive illustration of PPI networks over the entire duration of the maturation, transport, and assembly into virions of MuV glycoproteins. Indeed, the ESCRT-related factors (CHMP2B and CHMP4B), which are involved in the budding of MuV [21], were also identified using this method. However, the possibility that another sequence additional to the cytoplasmic region of MuV F and HN proteins may affect the intracellular transport of these proteins should be considered when interpreting these data. Indeed, the F-FLAG-Tb(C) and Tb(N)-HA-HN proteins were detected throughout the cytoplasm and the perinuclear region as well as in the PM, whereas the wildtype HN protein was expressed mainly in the PM. Therefore, the factors categorized as "nuclear-related protein" by GO analysis in particular need to be considered regarding this point.

Most membrane proteins are transported though the secretory pathway from the ER to the Golgi, from which they subsequently move to the TGN and finally to the PM [26]. SNARE proteins are central mediators of vesicular fusion events and thus play central roles in intracellular membrane trafficking [23]. They are classified into four groups, R-SNARE and Qa-, Qb-, and Qc-SNAREs, according to the central residue in their SNARE motifs. USE1 was first described in yeast as a Qc-SNARE protein required for COPI-mediated Golgi–ER retrograde transport [27]. Later, USE1 was reported to also be involved in the regulation of post-Golgi transport [28]. In addition, COPI proteins are essential for Golgi cisternal maturation and dynamics [29]. These findings indicate that USE1 plays critical roles in the regulation of membrane traffic between the ER and post-Golgi. The co-localization of USE1 with MuV F protein in the ER suggests that USE1 has a role in the maturation process of F protein in the early secretory pathway. In 293T and HeLa cells, the N-linked glycosylation of F protein was impaired by USE1 knockdown, indicating that USE1 is involved in the F protein trafficking that impacts glycosylation, potentially by increasing or decreasing the resident time of the F protein in a portion of the Golgi, or in trafficking cellular glycan-modifying enzymes. Notably, the N-linked glycosylation of the cellular protein integrin β1 was also impaired in USE1-knockdown cells. Therefore, USE1 is thought to be a broadly-important host component, rather than one specifically involved in MuV F maturation.

Although specific N-linked glycans on the F protein are important for the F protein functions of paramyxoviruses, there are no clear conserved roles for N-linked sites among these viruses. For example, mutation of the N-linked site in the heptad repeat region B (HRB) of parainfluenza virus 5 (genus *Orthorubulavirus*) F protein significantly reduced the intracellular

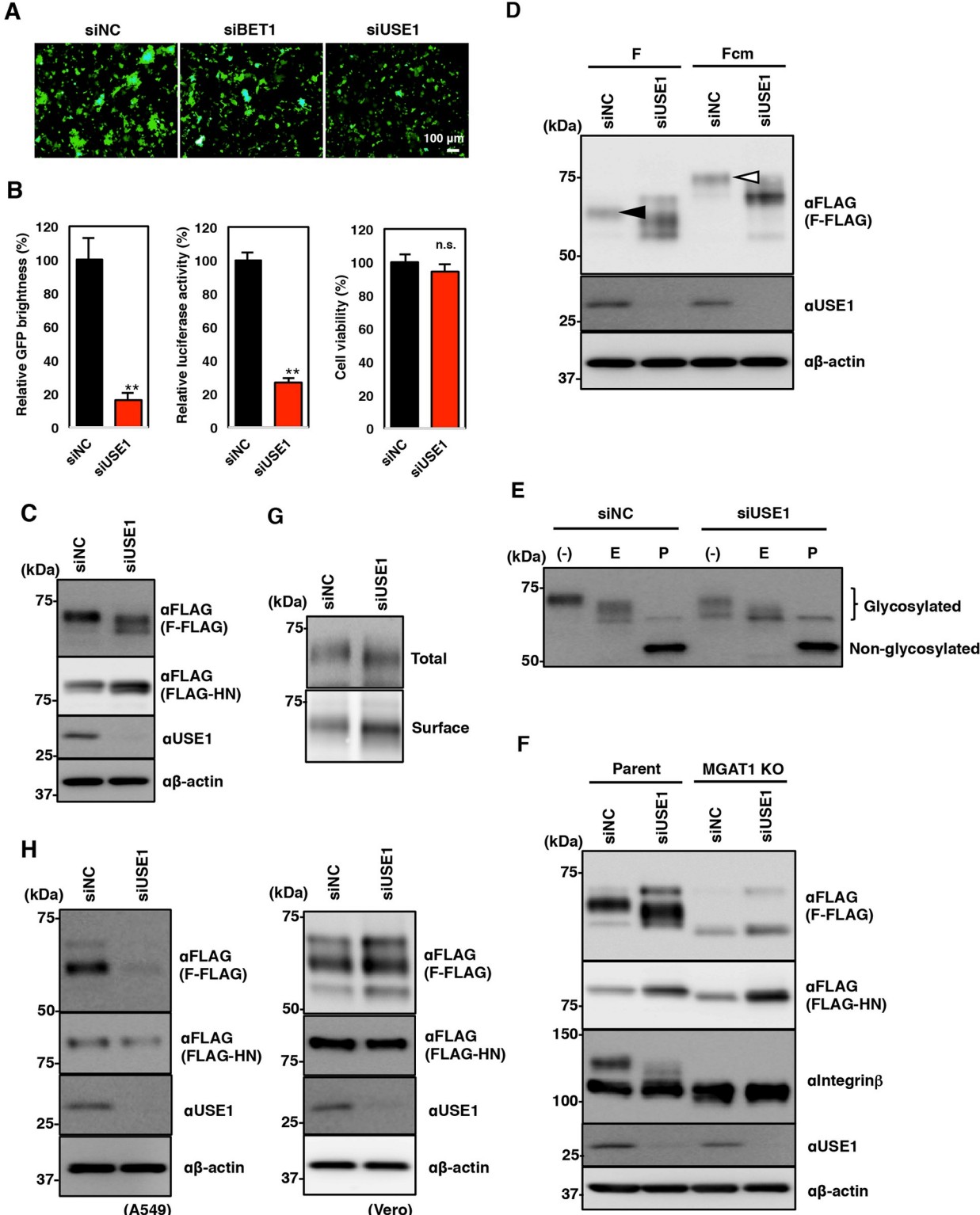

**Fig 6. Involvement of USE1 in the glycosylation and expression of MuV F protein. (A)** Immunofluorescence assay showing the syncytium formation induced by F and HN proteins in A549 cells transfected with siBET1, siUSE1, or siNC. **(B)** Relative AcGFP brightness, luciferase activity, and cell viability in USE-knockdown 293CD4/DSP$_{1-7}$ and 293FT/DSP$_{8-11}$ cells. Error bars indicate the standard deviations of triplicate wells. Differences with a *p*-value of <0.01(**) were considered to be statistically significant. (n.s. = not significant) **(C)** Immunoblotting showing the expression levels of F-FLAG, FLAG-HN, USE1, and β-actin in 293T cells transfected with siUSE1 or siNC. **(D)** Immunoblotting showing the

expression levels of F- and Fcm-FLAG, USE1, and β-actin in 293T cells transfected with siUSE1 or siNC. Black and white arrow heads indicate the bands corresponding to cleaved and uncleaved F proteins, respectively. **(E)** Immunoblotting showing the change in the mobility of the F protein bands in siUSE1- or siNC-transfected cells treated with Endo H (E) or PNGase F (P). **(F)** Immunoblotting showing the expression levels of F-FLAG, FLAG-HN, integrin β, USE1, and β-actin in HeLa/Parent or /MGAT1 KO cells transfected with siUSE1 or siNC. **(G)** Immunoblotting showing the total or surface expression of F-FLAG protein in 293T cells transfected with siUSE1 or siNC. **(H)** Immunoblotting showing the expression levels of F-FLAG, FLAG-HN, USE1, and β-actin in A549 or Vero cells transfected with siUSE1 or siNC. The experiments were performed at least three times, and representative data are shown.

transport and fusion activity of the protein [30], whereas removal of the N-linked glycan from a similar region of the F protein of NiV (genus *Henipavirus*) had little effect on the processing or fusogenicity of the protein [13]. An F protein of NDV (genus *Orthoavulavirus*) lacking the N-linked glycan of the corresponding site was expressed at nearly wildtype levels but had decreased fusion activity [10]. However, other paramyxoviruses, such as MeV and SeV (genus *Morbillivirus* and *Respirovirus*, respectively), do not contain an N-linked glycosylation site in this region [11,12,17,31]. In addition, glycosylation profiles (high-mannose, hybrid, or complex) differ depending on the site and virus species [32–34]. Therefore, it is necessary not only to identify the functional glycans of paramyxovirus F proteins but also to analyze the maturity of each glycan. However, there is limited available information on the functional roles of N-linked sites on the MuV F protein. Further investigation is required to clarify the functions of each glycan on the MuV F protein, taking into account their glycosylation profiles.

Although the detailed molecular mechanism for the repressed expression of F protein observed in USE1-knockdown A549 cells is unknown, one possibility is that an abnormal resident time of F protein in the ER and Golgi may have affected the F protein stability. Alternatively, because glycosylation is involved in protein stability, it is possible that the F protein was degraded by the lack of proper glycosylation, as in 293T and HeLa cells. In any case, the low expression of the F protein would have led to the inhibition of syncytium formation and the growth suppression present in USE1-knockdown A549 cells. In contrast, in Vero cells, USE1 was hardly involved in the glycosylation and expression of F protein, which may be the reason why a knockdown of USE1 in Vero cells was less effective in suppressing virus growth than it was in A549 cells. The altered glycosylation of the F protein in USE1-depleted 293T cells could be attributed to the reduced fusogenicity observed in the DSP-based fusion assay. However, it is difficult to provide direct evidence that immaturity of N-linked glycosylation of F protein led to a reduced level of membrane fusion activity, although N-glycans on paramyxovirus F proteins are known to be important for their fusion activity. Further investigation into the involvement of the glycosylation of F protein in viral propagation is also needed. Although the effect of USE1 knockdown on the F protein varies among different types of cultured cells, USE1 is still an important factor in the acquisition of F protein function.

Glycosylation is very diverse across species, tissues, and cells. Because MuV targets a variety of tissues, further *in vitro* and *in vivo* studies that more closely resemble actual infection conditions are necessary to evaluate the relationship between glycosylation of the F protein and viral growth and pathogenesis. However, this new identification of a host factor required for the proper glycosylation and the expression of MuV F protein should shed light on the molecular mechanisms underlying the maturation of paramyxovirus F protein and advance our understanding paramyxovirus propagation.

## Materials & methods

### Cell culture, viruses, and transfection

Vero (African green monkey kidney) cells were provided by the U.S. Food and Drug Administration and were maintained in Dulbecco's modified Eagle's medium (DMEM) (Nacalai

Tesque, Kyoto, Japan) supplemented with 100 U/ml penicillin, 100 mg/ml streptomycin (P/S), and 5% fetal bovine serum (FBS). A549 (human lung epithelial) (ATCC CCL-185) and 293T (human kidney) (ATCC CRL-3216) cells were purchased from the American Type Culture Collection (Manassas, VA, USA) and cultured in DMEM containing P/S and 10% FBS. HeLa-mCAT#8 clone (Parent) and HeLa-MGAT1 KO cells were generated as described previously and maintained in DMEM containing 10% FBS [25]. 293CD4/DSP$_{1-7}$ and 293FT/DSP$_{8-11}$ cells were generated as described previously [24] and maintained in DMEM containing 10% FBS and 1.5 μg/ml of puromycin. Cell viability was measured using Cell Count Reagent SF (Nacalai Tesque) in accordance with the manufacturer's instructions.

A wildtype MuV (Odate strain) [35] and a recombinant MuV expressing AcGFP (rMuV/AcGFP) [36] were used in this study, and infectious titers of virus were determined by plaque assay using Vero cells.

Cells were transfected with plasmid DNA using *Trans*IT-LT1 Transfection Reagent or *Trans*IT-293 Transfection Reagent (Mirus/TaKaRa Bio, Shiga, Japan) or with siRNA using Lipofectamine RNAiMax (Invitrogen/Thermo Fisher Scientific, Waltham, MA, USA) in accordance with the manufacturer's instructions. Commercially available siRNA pools targeting each cellular protein and a control non-targeting siRNA pool (siGENOME SMARTpool) were purchased from Dharmacon/Horizon Discovery (Lafayette, CO, USA).

## Plasmids

Plasmids pCAGGS-N, -M, -F, and -HN were generated as previously described [37,38]. C-terminally FLAG-tagged F and N-terminally FLAG-tagged HN were generated by PCR-based mutagenesis, resulting in the plasmids pCAGGS-F-FLAG and pCAGGS-FLAG-HN, respectively. A mutant of F-FLAG in which the Arg$^{99}$ and Lys$^{101}$ were replaced by Asn (Fcm-FLAG) was generated by PCR-based mutagenesis. C-terminally FLAG/split-TurboID$_{74-321}$(Tb(C))-fused F and N-terminally split-TurboID$_{2-73}$(Tb(N))/HA-fused HN were generated by PCR-based mutagenesis, resulting in the plasmids pCAGGS-F-FLAG-Tb(C) and pCAGGS-Tb(N)-HA-HN, respectively. The cDNAs encoding human BET1, EMD, LRRC59, USE1, or VAPA were each amplified from 293T cells by RT-PCR and cloned into pCAGGS for the expression of N- or C-terminally HA-tagged proteins, resulting in the plasmids pCAGGS-HA-BET1, -HA-EMD, -HA-LRRC59, -HA-USE1, and -VAPA-HA, respectively. Sequences of all plasmids were confirmed using an ABI PRISM 3500*xL* Genetic Analyzer (Applied Biosystems/Thermo Fisher Scientific).

## Antibodies

The anti-MuV M (79D), anti-MuV F (44C), and anti-MuV HN (78) mouse monoclonal antibodies (MAbs) and anti-MuV N and anti-MuV M rabbit polyclonal antibodies (PAbs) used here were described previously [39–41]. Anti-FLAG (M2) and anti-β-actin (AC-15) mouse MAbs were purchased from Sigma (St. Louis, MO, USA). Anti-HA (HA11) mouse MAb and HRP- and AF488-conjugated streptavidin were purchased from BioLegend (San Diego, CA, USA). Anti-Calnexin (MA3-027) mouse MAb and anti-BET1 (PA5-88961) rabbit PAb were purchased from Thermo Fisher Scientific. Anti-integrin β (ab183666) and anti-Calreticulin (ab2907) rabbit PAbs were purchased from Abcam (Cambridge, UK). Anti-USE1 (25218-1-AP) rabbit PAb was purchased from Proteintech (Rosemont, IL, USA).

## Cell extracts, immunoblotting, and immunoprecipitation

For the preparation of cell extracts, cells were washed twice with cold phosphate-buffered saline (PBS) and then lysed in cell lysis buffer (20 mM Tris-HCl, pH 7.5, 135 mM NaCl, 1% Triton-X 100, and protease inhibitor cocktail [Complete Mini; Roche, Mannheim, Germany]).

For immunoblotting, each cell lysate was boiled in SDS sample buffer and subjected to SDS-PAGE. The proteins were transferred to polyvinylidene difluoride membranes (Millipore, Bedford, MA, USA) and incubated with the appropriate antibodies. Each protein was visualized using SuperSignal West Femto Maximum Sensitivity Substrate (Thermo Fisher Scientific) and detected using an LAS-3000 image analyzer system (Fuji Film, Tokyo, Japan). For immunoprecipitation, each cell lysate was pre-cleaned with protein G–sepharose (GE Healthcare, Chicago, IL, USA). Antibody–protein complexes were purified with protein G beads and washed three times with cell lysis buffer. After being boiled in SDS sample buffer, the proteins were separated by SDS-PAGE and processed for immunoblotting.

### Endoglycosidase digestion

Immunoprecipitates were incubated at 100˚C for 10 min in denaturation buffer (New England Biolabs (NEB), Ipswich, MA, USA). After the protein G beads were removed by centrifugation, each sample was split into three equal aliquots. The first aliquot was used as an untreated control, and the second and the third aliquots were incubated with 500 U of Endo H (NEB) and PNGase F (NEB), respectively, at 37˚C for 1 h.

### Cell surface biotinylation

Cells were washed three times with cold PBS (pH 8.0) and then treated with 3 mM EZ-Link Sulfo-NHS-Biotin (Thermo Fisher Scientific) in cold PBS (pH 8.0) for 30 min at room temperature. The cells were washed three times in 100 mM glycine in cold PBS (pH 8.0) to remove the excess biotin, lysed in cell lysis buffer, and processed for immunoprecipitation and immunoblotting.

### Biotin labeling with split-TurboID

For biotin labeling of transfected cells, the cells were treated with 50 μM biotin (Nacalai Tesque) for 12 h. The cells were then washed twice with cold PBS, lysed in cell lysis buffer, and processed for immunoprecipitation and immunoblotting. For enrichment of biotinylated proteins, the cell lysate was incubated overnight with streptavidin-coated magnetic beads (Dynabeads MyOne Streptavidin C1, Thermo Fisher Scientific) on a rotator at 4˚C. The beads were then washed with cell lysis buffer three times and processed for immunoblotting.

### Mass spectrometry

For mass spectrometry analysis, duplicate samples of 293T cells were transfected with four plasmids (pCAGGS-N/M and -F-FLAG-Tb(C)/Tb(N)-HA-HN or -F/HN as control) and labeled with 50 μM biotin for 24 h. The cells were lysed with lysis buffer 1 (50 mM Tris-HCl [pH 7.4], 150 mM NaCl, 1% NP-40, 0.5% deoxycholate [DOC], 2 mM $MgCl_2$, 20 U benzonase, and protease inhibitor cocktail). After being incubated at 4˚C for 5 min, the cell lysate was mixed with an equal volume of lysis buffer 2 (50 mM Tris-HCl [pH 7.4], 150 mM NaCl, 1% NP-40, 0.5% DOC, 2 mM TCEP, 0.2% SDS, and 2 mM EDTA) and further incubated at 4˚C for 5 min. After the cell debris was removed by centrifugation at 15,000 rpm for 10 min at 4˚C, the resulting supernatant was immunoprecipitated using NanoLink streptavidin magnetic beads (TriLink, San Diego, CA, USA). The beads were washed with wash buffers 1 to 4 in order (wash buffer 1: 50 mM Tris-HCl [pH 7.4] and 2% SDS; wash buffer 2: 20 mM Tris-HCl [pH 7.4], 500 mM NaCl, 1% Triton-X100, 0.1% DOC, and 1 mM EDTA; wash buffer 3: 10 mM Tris-HCl [pH 7.4], 250 mM LiCl, 1% NP-40, 0.1% DOC, and 1 mM EDTA; and wash buffer 4: 50 mM Tris-HCl [pH 7.4] and 0.1% Triton-X100). The beads were then incubated at

100˚C for 5 min in elution buffer (5 mM biotin, 20 mM Tris-HCl [pH 7.4], 150 mM NaCl, 2% SDS, and 5 mM TCEP). A final concentration of 10 mM iodoacetamide was added to the eluted sample, and alkylation treatment was performed at room temperature for 30 min. The biotinylated proteins were precipitated with acetone, and the resulting precipitates were re-dissolved in digestion buffer (100 mM ammonium bicarbonate, 0.05% decyl β-D-glucopyranoside, and 7 M guanidine hydrochloride) and treated with lysyl endopeptidase (Fuji Film Wako Chemicals, Osaka, Japan) at 37˚C for 3 h. These samples were further digested with Trypsin TPCK (Sigma) at 37˚C for 12 h and then centrifuged at 15,000 rpm for 1 min. The resulting supernatant was subjected to liquid chromatography-tandem mass spectrometry (LC-MS/MS) analysis. MS analysis was performed twice per sample. For LC-MS/MS analyses, an Easy-nLC1200 system (Thermo Scientific) and Q Exactive HF-X (Thermo Scientific) were used. A C18 silica resin-packed capillary column with a diameter of 10 μM and a length of 12 cm (Nikkyo Technos) was used as the analytical column for LC. Solvents A (0.1% formic acid) and B (0.1% formic acid/80% acetonitrile) were used; peptide separation was performed at a flow rate of 300 nL/min with a gradient of B from 0% to 40% for 80 min.

For MS and MS/MS measurement, the Data-Dependent Acquisition (DDA) mode was used. The mass resolution was 60,000 for MS and 15,000 for MS/MS, and the m/z measurement range was 380–1,500 for MS and 200–2000 for MS/MS. The AGC target and Maximum IT were set to $3e^6$ and 60 ms for MS, and $1e^5$ and 45 ms for MS/MS, respectively. The MS/MS scan selected 20 precursor ions per MS scan with an exclusion time of 12 s.

MS/MS data sets were analyzed by Proteome Discoverer 2.2 software (Thermo Fisher Scientific). Peptide identification was performed using the SEQUEST algorithm with Swiss-Prot (Human: 20,386 entries) as the protein database. The tolerances were specified as 10 ppm and 0.02 Da for precursor and fragment ions, respectively. Trypsin was selected as the digestion enzyme, and up to two missed cleavages were allowed. Oxidation (M), biotinylation (K), carbamidomethylation (C), and protein N-terminal acetylation were added as modifications. For protein quantification, a label-free quantification method using precursor ions was used.

## GO enrichment and clustering analysis

Functional enrichment analysis of the host factors identified via mass spectrometry was performed using Database for Annotation, Visualization and Integrated Discovery (DAVID) [42]. The GO terms were classified into three categories: biological process, molecular function, and cellular component. GO terms with an RFDR of <0.05 and fold enrichment of >5 were selected for use in subsequent analyses. The functional annotation clustering analysis was performed by the gplot (https://CRAN.R-project.org/package=gplots), amap (https://CRAN.R-project.org/package=amap), and proxy (https://CRAN.R-project.org/package=proxy) packages in the R software with the Jaccard index and ward.D2 clustering method.

## Construction of PPI networks

The PPI networks of factors extracted from the GO analysis were constructed using the STRING (https://string-db.org) database [43]. A confidence score of >0.9 was used as the cutoff criterion. The visualization of PPI networks was performed by Cytoscape software using the Edge-weighted Spring Embedded Layout algorithm [44].

## siRNA screening

A549 cells seeded in 96-well plates were transfected with one of the experimental siRNAs. After 48 h, the cells were infected with rMuV/AcGFP at an MOI of 0.05. At 96 hpi, the AcGFP brightness was analyzed using a BZ-X800 analyzer (Keyence Co., Osaka, Japan).

## Quantitative RT-PCR (qRT-PCR)

Total RNA was extracted using the RNeasy Mini Kit (Qiagen, Hilden, Germany), and first-strand cDNA was synthesized using PrimeScript RTase (Takara). The reverse transcription reaction was performed using the following primers: 5'-ACCAAGGGGAAAATGGAGATG-3' (complementary to nucleotides 1 to 21 of the MuV genome) for MuV genomic RNA and an oligo(dT) primer for viral and cellular mRNAs. The amount of each target cDNA was measured using the Universal ProbeLibrary and the LightCycler 480 system (Roche) in accordance with the manufacturer's instructions. The levels of each target RNA were normalized to that of hypoxanthine phosphoribosyltransferase 1 (HPRT1) mRNA. The qPCR for the MuV N and HPRT1 genes was performed as described previously [36].

## Immunofluorescence microscopy

Cells were fixed in PBS containing 4% paraformaldehyde for 15 min at room temperature. The cells were then permeabilized with PBS containing 0.2% Triton X-100 for 10 min, blocked with PBS containing 2% bovine serum albumin for 30 min at room temperature, and incubated with the indicated antibodies. Nuclei were stained with 4', 6-diamidino-2-phenylindole (DAPI). Samples were examined under a FV3000 confocal laser-scanning microscope (Olympus, Tokyo, Japan).

## VLP assay

The plasmids pCAGGS-F and -HN or pCAGGS-F-FLAG-Tb(C) and -Tb(N)-HA-HN were introduced into 293T cells along with the plasmids pCAGGS-N and -M. The culture media were harvested at 48 h post-transfection and centrifuged at 7,500 x $g$ for 2 min to remove cell debris. The resulting supernatants were layered onto 20% sucrose in TNE (0.1 M NaCl, 0.01 M Tris-HCl [pH 7.5], and 0.001 M EDTA) (w/v) and centrifuged in centrifuge with an SW41 rotor (Beckman Coulter, Brea, CA, USA) at 140,000 x $g$ for 1.5 h. The resulting pellets were suspended in SDS sample buffer and processed for immunoblotting.

## DSP-based fusion assay

A DSP assay was performed as described previously [24]. A fusion protein of RL and GFP was split into two fragments designated as dual split proteins $DSP_{1-7}$ and $DSP_{8-11}$. Although each individual fragment lacks the activity of either RL or GFP, when these fragments are expressed simultaneously within the same cell, they reassemble and become functional. $293CD4/DSP_{1-7}$ and $293FT/DSP_{8-11}$ cells constitutively expressing $DSP_{1-7}$ and $DSP_{8-11}$, respectively, were mixed and cultured together and then transfected with pCAGGS-F and pCAGGS-HN. At 48 h post-transfection, the GFP brightness was analyzed using a BZ-X800 analyzer, and the RL activity was measured using the Renilla Luciferase Assay System (Promega, Madison, WI, USA) and a GloMax 20/20 Luminometer (Promega).

## Statistical analysis

Differences between groups were evaluated using unpaired Student's $t$-tests. Error bars indicate the standard deviations of triplicate measurements. Statistical significance was assumed at $^*p < 0.05$ and $^{**}p < 0.01$; n.s. = not significant.

## Data deposition

The raw mass spectrometry proteomics data have been deposited to the ProteomeXchange Consortium via the PRIDE partner repository with the dataset identifier PXD035151.

## Supporting information

**S1 Fig. Effect of introduction of TurboID into the C-terminus of M protein on VLP production.** VLP assay showing the amounts of the wildtype M or the M/TurboID proteins expressed in the cells (Lysate) or released into the supernatants (Sup).
(TIF)

**S2 Fig. Experimental design for MS-based proteomics.** 293T cells transfected with the indicated plasmids were labeled with 50 μM biotin. The cells were lysed, and biotinylated proteins were enriched using streptavidin beads, digested into peptides, and analyzed by LS-MS/MS. MS analysis was performed twice per sample. The averages of the detection intensities (protein abundances) obtained from each sample (control and TurboID) were calculated. Factors that were detected only in the split-TurboID sample or factors with an intensity ratio (TurboID/control) of $\geq$10-fold were then extracted, and a total of 641 proteins were identified as being significantly enriched in the split-TurboID-expressing cells.
(TIF)

**S3 Fig. Interaction of MuV F or HN protein with host proteins.** Immunoprecipitation assay showing the interaction of F-FLAG or FLAG-HN proteins with HA-EMD (left), HA-LRRC59 (center), or VAPA-HA (right) in 293T cells.
(TIF)

**S4 Fig. Intracellular localization of BET1 and USE1. (A)** Immunofluorescence of A549 cells treated with anti-Calnexin (red) and anti-BET (green) antibodies. (**B)** Immunofluorescence of A549 cells expressing HA-USE1 treated with anti-USE1 (red) and anti-HA (green) antibodies. (**C**) Immunofluorescence of A549 cells expressing HA-USE1 treated with anti-Calreticulin (red) and anti-HA (green) antibodies.
(TIF)

**S5 Fig. Color profile of immunofluorescence assay images.** (A–B) RGB line profiles along the lines shown in Fig 4B (A) and 4C (B) were analyzed by using ImageJ software (Version: 2.3.0).
(TIF)

**S1 Table. List of proteins identified to be enriched in the split-TurboID-expressed cells.**
(PDF)

## Acknowledgments

We acknowledge Zen Matsuda of The University of Tokyo for providing 293CD4/DSP$_{1-7}$ and 293FT/DSP$_{8-11}$ cells. We thank Tomohisa Hatta of Robotic Biology Institute for support for mass spectrometry analysis. We also thank all the members of the Department of Virology III, NIID, for their technical advice and critical input. We thank Katie Oakley, PhD, from Edanz (https://jp.edanz.com/ac) for editing a draft of this manuscript.

## Author Contributions

**Conceptualization:** Hiroshi Katoh, Makoto Takeda.

**Formal analysis:** Tsuyoshi Sekizuka.

**Funding acquisition:** Hiroshi Katoh.

**Investigation:** Yaqing Liu, Hiroshi Katoh, Chaewon Bae, Aika Wakata, Fumihiro Kato.

**Methodology:** Hiroshi Katoh, Fumihiro Kato, Masafumi Sakata, Toshiyuki Yamaji.

**Supervision:** Zhiyu Wang, Makoto Takeda.

**Writing – original draft:** Hiroshi Katoh, Makoto Takeda.

**Writing – review & editing:** Yaqing Liu, Tsuyoshi Sekizuka, Chaewon Bae, Aika Wakata, Fumihiro Kato, Masafumi Sakata, Toshiyuki Yamaji, Zhiyu Wang.

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
