## [Decision Letter · Decision Letter 0]

3 Aug 2022

Dear Dr. Katoh,

Thank you very much for submitting your manuscript "SNARE protein USE1 is essential for the glycosylation of mumps virus fusion protein and viral propagation" for consideration at PLOS Pathogens. As with all papers reviewed by the journal, your manuscript was reviewed by members of the editorial board and by several independent reviewers. In light of the reviews (below this email), we would like to invite the resubmission of a significantly-revised version that takes into account the reviewers' comments.

All three reviewers agreed that the use of spilt TurboID in a proximity ligation assay is a useful application to identify host interacting factors of putative biological significance. However, significant concerns were raised about data presentation and interpretation. Please pay specific attention to address data short-comings and clarity.  Specifically, Reviewer 3 raised valid concerns about over-interpretation and incomplete/incorrect interpretation regarding the cell biological function of USE1 (e.g., whether the effects on glycosylation are indirect as a result of its function (or lack thereof in KD cells) in Golgi to ER retrograde transport as opposed to direct effects on glycosyation per se). Please pay attention to these various interpretation issues.

In addition, Reviewer 1 raised valid concerns regarding current standards (& best practices) for publication of studies using Mass Spec ID.  While it might not be possible to redo the experiments in triplicates using the controls that have been suggested, please provide the additional information requested and any additional analyses that will help readers to understand the power AND limitations of the methods used. Note that standard practice requires deposition of raw Mass Spec data in repositories such as PRIDE Archive, MassIVE, jPOST and iProX (see ProteomeXchange consortium). 

We cannot make any decision about publication until we have seen the revised manuscript and your response to the reviewers' comments. Your revised manuscript is also likely to be sent to reviewers for further evaluation.

Sincerely,

Benhur Lee

Section Editor

PLOS Pathogens

Benhur Lee

Section Editor

PLOS Pathogens

Kasturi Haldar

Editor-in-Chief

PLOS Pathogens

orcid.org/0000-0001-5065-158X

Michael Malim

Editor-in-Chief

PLOS Pathogens

orcid.org/0000-0002-7699-2064

Reviewer's Responses to Questions

**Part I - Summary**

Reviewer #1: This study by Liu and co-authors describes a proximity labeling approach to identify protein in proximity to the MuV F/HN glycoproteins in HEK293T cells. Based on the findings from proteomics, the authors chose a subset to test 24 genes in an siRNA screen for MuV infection in A549 cells. The siRNA screen revealed 13 "hits" that were replicated in A549 and also tested in Vero cells. They focus on SNARE proteins BET1 and USE1 to validate physical interactions with F/HN by co-IP and co-localization studies and test their impact on infection, syncytia formation, and F glycosylation patterns.

The major strengths of the manuscript are the findings regarding USE1 and its effects on MuV infection, syncytia formation, and F glycosylation are novel and significant. The authors also demonstrate that USE1 is involved in host protein glycosylation (integral B1). These findings can help understand the basic biology of MuV glycosylation requirements for infection and syncytia formation.

Reviewer #2: In this study, Liu and colleagues have used a proximity labeling approach to identify host factors associated with the mumps virus (MuV) glycoproteins F and HN in transfected cells. Several proteins known to be involved in vesicle transport were identified, and of these, 13 were required for efficient replication of MuV in siRNA experiments. Two of these, USE1 and BET1, both SNARE proteins that function in ER/Golgi vesicle transport, were studied further, and USE1 in particular was found to be important for proper F protein N-linked glycosylation.

This work is quite novel - prior to this very little was known about host factor recruitment through MuV glycoproteins. The TurboID approach used here is powerful and appropriate for the questions being asked. In general, the data are clear and interpretation is sound.

Reviewer #3: While mumps virus is a long-standing cause of serious illness, much about its viral life cycle remains unknown. This manuscript seeks to elucidate additional host factors important for mumps infection by examination of proteins that are in close proximity to the viral glycoproteins. An interesting set of genes are identified, and the further studies suggest an impact from BET1 and USE1. Though the data as presented is solid, there are concerns about the interpretation of the data that somewhat limit enthusiasm.

**Part II – Major Issues: Key Experiments Required for Acceptance**

Reviewer #1: The major weaknesses of the manuscript relate to the description and application of proximity labeling proteomics that led to identification of USE1. The authors provide no negative control for the proximity labeling experiment. Proximity labeling technology is known to yield very high background, and that seems to be the case here considering the authors identified 638 protein in this analysis that almost certainly do not all interact with F/HN. At the very least, a control experiment where cells not supplemented with biotin were subjected to the same strep pulldown and MS analysis would allow subtraction of proteins non-specifically bindings to strep beads. Even better would be to provide a control experiment with half of turboID fused to a protein that does not interact with F/HN (GFP, for example). These experiments should be performed with at least three biological replicates, but it appears that these experiments were not replicated at all. Including replicates and negative controls will enable the authors to perform a statistical protein interaction analysis such as SAINT to calculate an interaction score for each protein based on the peptide counts or intensity values in the test samples compared to negative controls. These scores are extremely useful to rank the data and quickly identify the top hits instead of providing an unranked list of 638 proteins.

The description of the proteomics analysis is not acceptable for publication. The methods section merely states that samples were “subjected to liquid chromatography-tandem mass spectrometry analysis”. This section should provide a description of the instrument used, a detailed description of the acquisition method, and a detailed description of the settings used for Proteome Discoverer for proteomics data analysis. Table S2 containing the proteomics results should provide metrics of protein identifications such as the number of peptides identified per protein and the label-free quantification intensity of the proteins. Finally, it is standard practice for authors to deposit raw mass spectrometry data in a repository such as PRIDE.

Reviewer #2: (No Response)

Reviewer #3: 1. The authors suggest that the detailed molecular mechanisms of N-linked glycosylation of paramyxovirus F proteins is poorly understood – but this seems like an overstatement. N-linked glycosylation is a highly conserved process for all proteins receiving these modifications, and a large set of the host enzymes involved in this process has been identified. Numerous studies have examined the effect of glycan mutants in paramyxovirus fusion proteins on changes in folding, trafficking, etc. This statement should be modified to better acknowledge the true state of the field.

2. There is a very large difference in the effect of siRNA knock-down between A549 and Vero cells for a number of the factors tested in Figure 3, including for USE1, that is the focus of the paper, though the authors do not really comment on this. Why might the effect be so different between the two cells lines? And if USE1 is so critical for mumps infection – why is the effect so small in Vero cells?

3. In Figures 4B and 4C, the authors look a localization for F compared to BET1 and USE1. Why was the first examination done at 12 hours post infection? Given that F much utilize the secretory pathway for expression, modification and trafficking, it would be expected that it would be present in those regions much earlier in the infection process. In addition, while there is certainly close localization of F with these proteins, in most cases the signal appears to be close together, rather than truly co-localized, as there appears to be very little yellow signal in the merge. Quantitative assessment of co-localization would be helpful in better assessing this. Finally – the cellular location where the signals are close to each other is important but not commented on. BET1 and USE1 are ER/Golgi proteins, and that appears consistent with their localization in these images – suggesting that they have a role during F folding and trafficking in the early secretory pathway, but not a role later on in viral budding. The authors should more carefully examine their images and discuss these results in more depth.

4. It would be very helpful to put the viral protein level figure (currently Figure S3A-C) into the main text, potentially in Figure 4, as this is likely the reason why viral titers eventually reduce.

5. The overall reduction in viral titer is modest for BET1 and greater for USE1 in Figure 4E, but in both cases there is viral propagation, though the title suggests USE 1 is “essential” for viral propagation. The title should be revisited to more accurately reflect the data, and the discussion should acknowledge this.

6. The data with integrin � in Figure 5E indicates that USE1 plays a role in complex carbohydrate addition to more membrane proteins than just mumps F, likely reflecting a key role for USE1 in trafficking of either the proteins that are modified or trafficking of components involved in glycan modification. Thus, USE1 appears to be a broadly-important host component, rather than one specifically involved in mumps F maturation – but this is not really reflected in the discussion.

7. The modification of N-glycans is cell type dependent, so it is important that the findings be examined in more than one cell type – especially as the effect of USE1 knock-down varied between A549 and Vero cells.

8. The authors suggest that USE1 is the first host factor identified as being involved in glycosylation of a paramyxovirus F protein – but this is completely incorrect. As stated earlier, N-linked glycosylation is a highly conserved process involving a set of enzymes in the ER which add the first high-mannose carbohydrate to the site. Further modifications are then carried out later by other identified host proteins. All of these factors are involved in F protein N-linked carbohydrate modification. In fact, analysis of viral glycoprotein modifications was a key part of identifying all of these components.

In addition, the change observed does not verify that USE1 actually is involved in glycosylation - or more accurately modification of the glycan chains. It suggests that USE1 is playing a role in trafficking (consistently with its cellular function) that impacts modification of the carbohydrate chains, potentially by increasing or decreasing the resident time of mumps F in a portion of the Golgi, or in trafficking cellular glycan-modifying enzymes. In fact, given the cytosolic localization of USE1, it would not be available to directly act on the N-linked carbohydrates in mumps F.

**Part III – Minor Issues: Editorial and Data Presentation Modifications**

Reviewer #1: (No Response)

Reviewer #2: Some issues that can be addressed to improve the manuscript:

1. In the Introduction section, it would be helpful to provide the reader with some introduction to the split-TurboID approach that forms the basis for this study. I.e., how these proximity-based strategies work in general; advantages as compared to traditional affinity purifications; difference between BioID and Turbo-ID; etc.

2. Authors state that insertion of split-TurboID was tolerated, but reduced VLP production by about 10-fold. On the other hand, when the complete TurboID was added to N, M, or F, no VLPs could be produced. It would be helpful to show one of these negative VLP results as a control, to confirm that the 10-fold reduced VLP production observed in Fig. 1C is meaningful in comparison to this true negative, and thus represents more than background signal.

3. For Table S1, it is not clear how this list has been sorted or if the order has any meaning.

4. Fig. 3A. The 13 that showed significant decreases in AcGFP expression should somehow be indicated directly on the figure.

5. From authors’ brief description of Fig. 3B in the Results section, one could get the impression that the 13 candidates broke neatly into two categories - five that had significant effects on MuV replication, and eight that didn’t. In reality the data is more complex. In some cases there are striking differences between results obtained in A549 cells vs. Vero cells. Also, the magnitudes of viral replication defects vary significantly. This is tacitly acknowledged in that siLYPLA2 was not included in the five, as it results in only modest replication defects. But its exclusion makes authors’ statement starting on Line 158 untrue. This entire paragraph needs to be re-written to give readers a more complete description and understanding of the data, and to more accurately convey what the criteria were for selecting the five.

6. Fig. 4E is missing the legend that shows which colored line corresponds to which virus.

7. Fig. S3C shows clear differences in N protein accumulation after BET1 or USE1 knockdown. Quantification of these important results should be shown.

8. Throughout Fig. 5 when analyzing the F proteins with different mobilities on the blots, it should be made more clear whether what we are looking at is cleaved F or uncleaved F.

9. Please provide a brief description of the dual split protein assay system and how it works in the Results section for the benefit of readers who are unfamiliar with it.

10. Results and/or Figure legend 5 should briefly state what approach has been used to measure cell surface transport of F.

Reviewer #3: (No Response)

PLOS authors have the option to publish the peer review history of their article (what does this mean?). If published, this will include your full peer review and any attached files.

Reviewer #1: No

Reviewer #2: No

Reviewer #3: No
---

## [Editor Report · Decision Letter 1]

24 Oct 2022

Dear Dr. Katoh,

We are pleased to inform you that your manuscript 'SNARE protein USE1 is involved in the glycosylation and the expression of mumps virus fusion protein and important for viral propagation' has been provisionally accepted for publication in PLOS Pathogens.

Best regards,

Benhur Lee

Section Editor

PLOS Pathogens

Benhur Lee

Section Editor

PLOS Pathogens

Kasturi Haldar

Editor-in-Chief

PLOS Pathogens

orcid.org/0000-0001-5065-158X

Michael Malim

Editor-in-Chief

PLOS Pathogens

orcid.org/0000-0002-7699-2064
---

## [Editor Report · Acceptance letter]

2 Nov 2022

Dear Dr. Katoh,

We are delighted to inform you that your manuscript, "SNARE protein USE1 is involved in the glycosylation and the expression of mumps virus fusion protein and important for viral propagation," has been formally accepted for publication in PLOS Pathogens.

Best regards,

Kasturi Haldar

Editor-in-Chief

PLOS Pathogens

orcid.org/0000-0001-5065-158X

Michael Malim

Editor-in-Chief

PLOS Pathogens

orcid.org/0000-0002-7699-2064